# Time Course of Plasma Proteomic and Oxylipin Changes Induced by LPS Challenge and Modulated by Antioxidant Supplementation in a Randomized Controlled Trial

**DOI:** 10.3390/antiox14050536

**Published:** 2025-04-29

**Authors:** Gerhard Hagn, Andrea Bileck, Thomas Mohr, Doreen Schmidl, David M. Baron, Bernd Jilma, Leopold Schmetterer, Gerhard Garhöfer, Christopher Gerner

**Affiliations:** 1Department of Analytical Chemistry, Faculty of Chemistry, University of Vienna, Waehringer Straße 38, 1090 Vienna, Austria; gerhard.hagn@univie.ac.at (G.H.); andrea.bileck@univie.ac.at (A.B.); thomas.mohr@univie.ac.at (T.M.); 2Vienna Doctoral School in Chemistry (DoSChem), University of Vienna, Waehringer Straße 42, 1090 Vienna, Austria; 3Joint Metabolome Facility, University of Vienna and the Medical University of Vienna, Waehringer Straße 38, 1090 Vienna, Austria; 4Department of Clinical Pharmacology, Medical University of Vienna, 1090 Vienna, Austria; doreen.schmidl@meduniwien.ac.at (D.S.); bernd.jilma@univie.ac.at (B.J.); leopold.schmetterer@meduniwien.ac.at (L.S.); 5Department of Anaesthesia, Intensive Care Medicine and Pain Medicine, Clinical Division of General Anaesthesia and Intensive Care Medicine, Medical University of Vienna, 1090 Vienna, Austria; david.baron@meduniwien.ac.at; 6Singapore Eye Research Institute, Singapore National Eye Centre, Singapore 168751, Singapore; 7Ophthalmology and Visual Sciences Academic Clinical Program, Duke-NUS Medical School, Singapore 169857, Singapore; 8SERI-NTU Advanced Ocular Engineering (STANCE), Singapore 168751, Singapore; 9School of Chemistry, Chemical Engineering and Biotechnology, Nanyang Technological University, Singapore 637371, Singapore; 10Center for Medical Physics and Biomedical Engineering, Medical University of Vienna, 1090 Vienna, Austria; 11Institute of Molecular and Clinical Ophthalmology, 4031 Basel, Switzerland

**Keywords:** antioxidant supplementation, inflammation, Lands cycle, lipids, lipopolysaccharide, oxylipins, proteomics, resolution of inflammation

## Abstract

Systemic molecular responses to pathogen-associated molecular patterns and their modulation by antioxidants are poorly understood in humans. Here, we present a two-stage clinical interventional study in healthy humans challenged with lipopolysaccharide. In the first step, the kinetics of inflammatory modulators within 8 h were investigated by plasma proteomics and lipidomics. In a second step, the effects of a placebo-controlled antioxidant intervention on the individual responses prior to another lipopolysaccharide challenge were determined. Plasma proteomics revealed an early involvement of the endothelium and platelets, followed by the induction of liver-derived acute phase proteins and an innate immune cell response. Untargeted lipidomics revealed an early release of fatty acids and taurocholic acid, followed by complex regulatory events exerted by oxylipins. The consistent lipopolysaccharide-induced downregulation of lysophospholipids suggested the involvement of the Lands cycle, and the downregulation of deoxycholic acid reinforced emerging links between the inflammasome and bile acids. Groups of molecules with similar kinetics to lipopolysaccharide challenge were observed to share precursors, synthesizing enzymes or cellular origin. Dietary antioxidant supplementation prior to lipopolysaccharide challenge had no detectable effect on protein kinetics but significantly downregulated pro-inflammatory sphingosine-1-phosphate and increased levels of oxylipins, 20-HEPE, and 22-HDoHE, which have been described to facilitate the resolution of inflammation. The present study identified a complex network of lipid mediators deregulated in plasma upon lipopolysaccharide challenge and highlighted the role of platelets, endothelial cells, and erythrocytes as potential inflammatory modulators. While dietary antioxidant supplementation hardly affected the initiation of inflammation, it may exert its effects supporting the resolution of inflammation.

## 1. Introduction

While the molecular mechanisms underlying inflammatory processes at the cellular level are well understood, less is known about the regulatory mechanisms at the whole-organism level. Proteins and lipids are involved in most molecular processes and modulate inflammatory responses, potentially acting as mediators or inhibitors. We have previously observed that antioxidant supplementation is able to alleviate clinical symptoms induced by endotoxemia [1,2]. The present study was designed to investigate systemic changes in proteins and lipids following an inflammatory challenge, with the aim of learning more about the molecular mechanisms of antioxidant supplementation. A better understanding of these processes may help to improve clinical management of inflammatory conditions.

Indeed, the physiological consequences of inflammation include numerous changes in the proteome [3,4] the lipid metabolism [5], the formation of reactive oxygen species [6], and lipid peroxidation products [7]. Cytokines such as interleukin 1-beta and TNF-alpha are among the major transcriptional targets of inflammatory signaling via NF-kappaB, STATs family members, and MAP kinases [8,9] and are the most potent promoters of inflammation. In vivo, cytokines act in a systemic manner and can cause the escalation, or “cytokine storm”, that is characteristic of some acute disease states [10,11,12]. Therefore, cytokines are valuable targets for therapeutic intervention [13]. However, due to methodological problems, cytokines are rarely used as clinical markers. Rather, the systemic response to cytokine action is responsible for the formation of molecules used as markers, such as the acute-phase proteins CRP (C-reactive protein) and SAA1 produced by the liver, and calprotectin produced by innate immune cells [14,15,16]. The systemic distribution of these proteins in the body and a massive increase correlating with inflammation severity make them ideal clinical markers.

Because of their short half-lives, lipid mediators are primarily considered to be local modulators. Prostaglandins such as PGE2 are thought to be responsible for most of the clinical symptoms characteristic of inflammation, and pharmacological inhibition of the formation of prostaglandins and other oxylipins is the mode of action of the most widely used anti-inflammatory drugs [17,18]. Some studies have investigated plasma oxylipin changes in chronic diseases such as metabolic syndrome and diabetes [19]. Importantly, lipids are considered to represent critical factors that regulate the resolution of inflammation [20], the failure of which may characterize chronic inflammation [21]. Although it has been shown that the abundance and physiological effects of lipid mediators can also escalate during acute infection [22], these molecules are not yet used as markers in clinical practice.

The therapeutic modulation of immune functions by antioxidants has been demonstrated for many years [23,24,25] and has stimulated a profound discussion regarding the use of dietary supplementation [26]. However, whether and how inflammatory processes could be modulated by dietary nutritional supplementation with antioxidants, including polyunsaturated fatty acids (PUFAs), is still not fully understood in humans.

To address these questions on a systemic level, we used a well-established experimental model of endotoxemia induced by a lipopolysaccharide (LPS) challenge in a clinical setting [27,28,29,30]. The involvement of immune cells in the immediate inflammatory response has previously been demonstrated by real-time polymerase chain reaction (RT-PCR) measurements of cytokines within 2 h after the LPS challenge [31]. Importantly, the previous studies have demonstrated that antioxidant supplementation improved the recovery from the inflammatory challenge, as a reduced response of retinal vascular reactivity to systemic hyperoxia was effectively restored by 2 weeks of antioxidant supplementation [1,2]. However, the molecular mechanism of this intervention was not known.

We, therefore, performed molecular profiling based on mass spectrometry to obtain time courses of proteins and lipids altered by the inflammatory challenge. Time-course analysis requires synchronization of inflammation initiation, as achieved in the current experimental strategy, but allows us to detect rather subtle changes. We hypothesized that if the antioxidants affected lipids or proteins, the time course of the corresponding molecules would be affected. Furthermore, the identity of the proteins found to be altered by the inflammatory challenge could provide valuable information about the cellular origin, as has been previously demonstrated in clinical studies of ulcerative colitis [32] and long COVID [33] using the same methods. The application of this strategy resulted in the realization that blood components other than immune cells, such as erythrocytes and platelets, play an active role in the regulation of inflammatory processes and suggests that lipids are mainly affected by anti-oxidative supplementation.

## 2. Materials and Methods

### 2.1. Study Design

Subjects were recruited by the Department of Clinical Pharmacology at the Medical University of Vienna. The study was actually registered at ClinicalTrials.gov number NCT00914576, approved by the Ethics Committee of the Medical University of Vienna (EC No.: 64/2009), and conducted in accordance with the guidelines of the Helsinki Declaration. The study has been registered at ClinicalTrials.gov number NCT00914576. For this randomized, double-masked, placebo-controlled parallel group study, 30 healthy male human individuals were included upon passing a screening examination and signing written informed consent prior to study entry. Participants with any clinically relevant illness, intake of medication, including vitamin or mineral supplements, or blood donation within 3 weeks prior to the study were excluded. Individuals who did not complete the study were replaced. In addition, participants had to abstain from alcohol- or caffeine-containing beverages within 12 h prior to each study day of the dual-stage interventional trial, which was randomized and placebo-controlled during the second stage.

To induce systemic inflammation and oxidative stress, an intravenous infusion of a bolus containing 2 ng/kg bodyweight of *Escherichia coli* endotoxin, known as lipopolysaccharide (LPS; NIH-CC, Bethesda, MD, USA), was used. Whole blood was collected using EDTA-anticoagulated collection tubes at baseline (BL), 60 min, 120 min, 240 min, and 480 min after LPS infusion. Plasma was obtained immediately by centrifugation at 4 °C at 2000× *g* for 10 min. After centrifugation, all samples were immediately frozen in pre-labelled Eppendorf safe-lock tubes at −80 °C until analysis. After this first stage, participants were randomly assigned to take either the omega-3 fatty acid containing food supplement Vitamac (*n* = 17; Croma Pharma GmbH, Korneuburg, Austria) or matching lactose and wheat starch containing placebo capsules (*n* = 13) for 14 days. The ingredients of the food supplement are specified in Appendix A. After the 14-day intervention, all participants were challenged again with LPS in the second stage and plasma was collected at baseline (BL), 60 min, 120 min, 240 min, and 480 min after LPS infusion, applying the same protocol described for the first stage.

### 2.2. Plasma Proteomics

The use of blood plasma was preferred to serum in order to avoid confounding effects as described earlier [34]. For the untargeted plasma proteomics analysis, plasma samples were thawed on ice and diluted 1:20 in lysis buffer (8 M urea, 50 mM triethylammonium bicarbonate (TEAB), 5% sodium dodecyl sulfate (SDS)). After heating at 95 °C for 5 min, the protein concentration was determined using a BCA assay, and 20 µg of protein were digested, applying the ProtiFi S-trap technology [35]. Here, a reduction and carbamidomethylation of solubilized proteins was performed using dithiothreitol (DTT) and iodoacetamide (IAA). Samples were then loaded onto S-trap mini cartridges, washed, and digested using a Trypsin/Lys-C Mix (1:40 enzyme to substrate ratio) at 37 °C for 2 h. Upon digestion, the resulting peptides were eluted, dried, and stored at −20 °C until liquid chromatography—tandem mass spectrometry (LC-MS/MS) analysis.

For the LC-MS/MS analysis, samples were reconstituted in 5 µL of 30% formic acid (FA) containing synthetic standard peptides and diluted with 40 µL of loading solvent (97.9% H_2_O, 2% acetonitrile (ACN), 0.05% trifluoroacetic acid). The separation of peptides was achieved using a Dionex Ultimate3000 nanoLC-system (Thermo Fisher Scientific, Waltham, MA, USA) equipped with an analytical column (25 cm × 75 µm, 1.6 µm C18 Aurora Series emitter column, (Ionopticks, Collingwood, Australia)). The injection volume was 1 µL, the flow rate was 300 nL/min, and the total run time was 85 min, including column washing and equilibration. The gradient flow profile started at 7% and increased to 40% mobile phase B (79.9% ACN, 20% H_2_O, 0.1% FA) over 43 min. The nanoLC-system was coupled to the timsTOF Pro mass spectrometer (Bruker, Billerica, MA, USA) equipped with a captive spray ion source operating in Parallel Accumulation–Serial Fragmentation mode as previously described [33].

Data analysis, including protein identification and label-free quantification (LFQ), was performed using the MaxQuant software package version 1.6.17.0 running the Andromeda search engine [36]. Raw data were searched against the SwissProt database “homo sapiens” (version 141,219 with 20,380 entries), allowing a maximum of two missed cleavages, a peptide tolerance of 20 ppm, carbamidomethylation on cysteines as fixed modification, as well as methionine oxidation and N-terminal protein acetylation as variable modification. A minimum of one unique peptide per protein was required for positive identification, while the “match between runs” option and a false discovery rate (FDR) ≤ 0.01 was applied. Identified proteins were filtered for common contaminants and reversed sequences using the Perseus software package version 1.6.14.0 [37]. LFQ intensities were log2-transformed, filtered for their number of independent identifications in a minimum of 70% of samples in at least one group, and missing values were replaced from a normal distribution. Volcano plots were generated using the Perseus software package version 1.6.14.0, and the time course profile of proteins was plotted using GraphPad Prism version 6.07 (2015).

### 2.3. Plasma Lipidomics

LC-MS/MS analysis of lipid mediators was performed as described previously [33]. Briefly, plasma samples were thawed on ice, and 400 µL of plasma were then mixed with cold ethanol (EtOH; 1.6 mL, abs. 99%, −20 °C; AustrAlco, Spillern, Austria), including an internal oxylipin standard mixture (Cayman Chemical, Tallinn, Estonia). The exact concentrations of each internal standard can be found in Appendix A. Samples were stored at −20 °C overnight for protein precipitation. Solid-phase extraction (SPE) was performed using StrataX SPE columns (30 mg mL^−1^; Phenomenex, Torrance, CA, USA). The eluted samples were dried, reconstituted in 150 µL of reconstitution solvent (H_2_O:ACN:methanol (MeOH) + 0.2% FA–vol% 65:31.5:3.5), subsequently transferred into an autosampler held at 4 °C, and measured via LC-MS/MS.

For the untargeted LC-MS/MS analysis, lipid mediators were separated using a Thermo Scientific^TM^ Vanquish^TM^ (UHPLC) system equipped with a Kinetex^®^ XB-C18-column (2.6 μm XB-C18 100 Å, LC Column 150 × 2.1 mm; Phenomenex^®^). A gradient flow profile was applied starting at 35% mobile phase B (mobile phase A: H_2_O + 0.2% FA, mobile phase B: ACN:MeOH (vol% 90:10) + 0.2% FA), increasing to 90% B (1–10 min), further increasing to 99% B within 0.5 min, and held for 5 min. Solvent B was then decreased to the initial level of 35% within 0.5 min, and the column was equilibrated for 4 min, resulting in a total run time of 20 min. The flow rate was 0.2 mL/min, the injection volume was 20 µL, and the column temperature was 40 °C. All samples were measured in technical duplicates in negative ionization mode, whereas one injection was performed in positive ionization mode. For the mass spectrometric analysis, the Vanquish^TM^ (UHPLC) system was coupled to a Q Exactive^TM^ HF Quadrupole-Orbitrap^TM^ high-resolution mass spectrometer (Thermo Fisher Scientific^TM^, Vienna, Austria), equipped with an HESI source operating in negative and positive ionization modes. The MS scan range was set to 250–700 *m*/*z*, with a resolution of 60,000 (at *m*/*z* 200) on the MS1 level and 15,000 (at *m*/*z* 200) on the MS2 level. For the fragmentation, a Top 2 method (HCD 24 normalized collision energy) with an inclusion list was applied (Appendix A). The spray voltage was set to 3.5 kV in negative and positive ionization mode, and the capillary temperature to 253 °C. Sheath gas was set to 46 and auxiliary gas to 10 arbitrary units.

For the data analysis, analytes were compared to an in-house established compound database based on the exact mass and retention time on the MS1 level using the TraceFinder^TM^ software package version 4.1 (Thermo Fisher Scientific^TM^). In addition, MS/MS fragmentation spectra were compared to reference spectra of in-house measured, commercially available standards or to reference spectra from the LIPID MAPS depository library from March 2023 [38]. The degree of identification of all analytes is shown in Appendix A. Subsequently, relative quantification of the identified analytes was performed on the MS1 level using the TraceFinder^TM^ software package version 4.1. The resulting peak areas were loaded into the R software package environment version 4.2.0 [39], log2-transformed, and normalized to the internal standards. Therefore, the mean log2-transformed peak area of the internal standards was subtracted from the log2-transformed analyte peak area. To enable missing value imputation using the minProb function of the imputeLCMD package version 2.1, the log2-transformed normalized peak areas were increased by 20 [40]. Significant differences between the study cohorts were determined using a linear model and the empirical Bayes method implemented in the limma R package, and the resulting *p*-values were adjusted for multiple testing according to the Benjamini–Hochberg procedure [41,42]. Data visualization was performed using the ggplot2 R package [43].

### 2.4. 3 Data Sharing Statement

The mass spectrometry proteomics data have been deposited to the ProteomeXchange Consortium via the PRIDE [44] partner repository with the dataset identifier PXD051013.

Data derived from the lipid mediator analysis are available at the NIH Common Fund’s National Metabolomics Data Repository (NMDR) website, the Metabolomics Workbench, https://www.metabolomicsworkbench.org (accessed on 2 February 2025) [45], where they have been assigned to the following studies: Study ID ST003137 and ST003138. The DOI for this project (PR001950) is: http://dx.doi.org/10.21228/M8NX50.

## 3. Results

This clinical intervention study consisted of two stages. At stage 1, the inflammatory responses to an LPS intervention were investigated using proteome profiling and oxylipin analysis. The hypothesis that an antioxidative intervention may affect inflammatory responses was investigated at stage 2, employing a placebo-controlled study design (Figure 1).

### 3.1. Plasma Proteome Profiling Indicates the Involvement of Endothelial Cells, Platelets, Innate Immune Cells, and the Liver During LPS-Induced Inflammation

Plasma was collected from 30 individuals at baseline (BL) and 60 min, 120 min, 240 min, and 480 min after LPS challenge (Figure 1, stage 1). Label-free shotgun analysis identified a total of 226 proteins after filtering for high confidence (FDR < 0.01 at protein and peptide level, two peptide identifications per protein) and robustness (independent identification of each protein in at least 21 out of 30 samples (70%) per time point, Appendix A). Paired *t*-tests identified no significant event 60 min after LPS challenge, whereas the platelet- and endothelium-derived protein VWF [34,46] was found to have increased 2 h after the challenge. After another 2 h, the liver-derived biomarker for inflammation, SAA1, and the innate immune cell-derived alarmin, S100A9, were found to be upregulated. Only 8 h after the challenge, the acute-phase protein CRP was found to be upregulated, in addition to LRG1 (Figure 2).

### 3.2. Plasma Lipidomics Demonstrates Immediate Adaptive Response Involving Oxylipins, Bile Acids, and the Lands Cycle in Response to LPS Challenge

The oxylipin assay based on the MS analysis of small lipids identified 25 fatty acids, 53 oxylipins, 28 lysolipids, six bile acids, four endocannabinoids, cortisol, and sphingosine-1-phosphate in the human plasma samples. All molecules were identified reproducibly (independent identification in >60% of the members of at least one group) via exact mass (deviation < 2 ppm), the isotopic patterns, and the molecular fragment ions (MS2 spectra). Seventy-one identifications were verified using analytical standards (Appendix A). Another 18 molecules were identified as oxylipins based on the sum formula and fragment spectra. However, due to the structural ambiguity of oxylipin isobars, it was not yet possible to assign unambiguous structures to these features. These oxylipins have been designated here as “molecular mass_chromatographic retention time” (Appendix A).

Paired *t*-tests again identified no significant event 60 min after LPS challenge, whereas 10 different molecules were found to be significantly deregulated (adjusted *p*-value < 0.01, more than 2-fold regulated) after 120 min. Besides the polyunsaturated fatty acids stearidonic acid, alpha- and gamma-linolenic acid, and *trans*-isoforms thereof, also a *trans*-isoform of DHA, was found to be upregulated (Appendix A). The upregulation of the platelet-derived oxylipins TxB2 (significant but less than two-fold), 12-HHTrE, and 12-HETE as described previously [34] indicated an early involvement of platelets. The upregulation of taurocholic acid and a downregulation of cholic acid indicated an early involvement of the liver. After another 2 h, the uniform downregulation of various lysophospholipids was striking. The lysophospholipid LPC (0:0/18:2) exhibited a downregulation to approximately 53% of its initial value after 8 h (remaining below a 2-fold change), with an adjusted *p*-value better than 1E-126. Plasma lysophospholipid levels correspond to the Lands cycle, responsible for the remodeling of phospholipids upon oxidative stress [47]. The present data thus demonstrated a direct involvement of the Lands cycle in the systemic response to LPS. In addition, one of the best-characterized sphingolipids, sphingosine-1-phosphate, was significantly downregulated as well upon LPS challenge. This sphingolipid is a specific product of erythrocyte and platelet metabolism and an essential regulator of inflammatory signaling [48,49].

On the other hand, the anti-inflammatory PUFA-derived oxylipins 13-OxoODE, 20-HETE, 22-HDoHE, 20-HEPE were upregulated. Eight hours after the LPS challenge, most of these alterations remained detectable. Remarkably, the polyunsaturated fatty acids stearidonic acid, alpha- and gamma-linolenic acid, and *trans*-isoforms thereof were found downregulated at that time, demonstrating complex dynamics. This observation may indicate some rapid adaptive responses of the human organism, as the anti-inflammatory bile acids glycolithocholic acid, glycodeoxycholic acid, and glycocholic acid were found to be upregulated at this time point (Figure 3).

### 3.3. Dietary Antioxidant Supplementation Does Not Affect the Kinetics of Inflammatory Marker Proteins Within Eight Hours After LPS Challenge

After stage 1 of the analysis, the donors were divided into two groups and treated for 14 consecutive days either with a placebo (*n* = 13) or an antioxidative treatment scheme (verum, *n* = 17), as depicted in Figure 1. All participants were then again treated with LPS as described above to obtain a similar time series of plasma samples. All LPS-induced proteome alterations observed during the first part of the study were reproduced. However, no significant event was detected when comparing the antioxidant treatment group to the placebo group at any time. Depicting time courses of average values of selected proteins demonstrates the uniformity of the three groups (Figure 4). These observations suggested that the antioxidant intervention had no detectable effect on the expression of inflammatory marker proteins.

### 3.4. Dietary Antioxidant Supplementation Increases Specialized Pro-Resolving Lipid Mediators and Downregulates Sphingosine-1-Phosphate After the LPS Challenge

The antioxidative intervention, lasting for 14 days, resulted in significant alterations of the plasma oxylipin composition: the PPAR-gamma agonist 15-deoxy-PGJ2 was found downregulated, whereas the anti-inflammatory 20-HEPE was found upregulated (Figure 5A, baseline, Appendix A). This may be remarkable, as 20-HEPE was found at stage 1 to be upregulated during the LPS intervention.

Almost all LPS-induced alterations of oxylipins observed at stage 1 were reproduced at stage 2, with only minor differences between the two groups. Only in the antioxidant group, 2 h after LPS, the anti-inflammatory DHA-derived oxylipin 22-HDoHE was found upregulated, whereas the neutrophil-derived catabolic product of the pro-inflammatory leukotriene LTB4, 6trans-LTB4, was found downregulated (Figure 5A). Sphingosine-1-phosphate, a specific product of erythrocyte, endothelial, and platelet metabolism [50], was consistently downregulated in the antioxidant treatment group, with a significant difference 4 h after LPS challenge (Figure 5B).

### 3.5. Similar Kinetics After LPS Challenge Suggests Common Biosynthetic Pathways

The stable time courses observed for most oxylipins showed clock-like features and allowed us to define eight distinct groups displaying specific kinetic features upon LPS challenge: (1) continuously upregulated (*n* = 3); (2) continuously upregulated, then partially down (*n* = 18); (3) first upregulated, then down below baseline levels (*n* = 23); (4) upregulated, down and up again (*n* = 1); (5) continuously downregulated (*n* = 10); (6) continuously downregulated then partially up (*n* = 24); (7) first downregulated, then up above baseline levels (*n* = 9); and (8) downregulated, up and down again (*n* = 35). Figure 6 displays representative examples of each category. These groups were found to be populated by molecules sharing a common precursor or a common enzyme responsible for their synthesis (Appendix A), thus potentially representing functional categories. Category 1 contained the endocannabinoid docosahexaenoyl ethanolamide (DHA-EA) and other ethanolamides. Category 2 contained mainly oxylipins described as being formed by platelets. Mainly polyunsaturated fatty acids were found in category 3. Category 5 contained many lysophosphatidylcholines, whereas category 6 mainly contained lysophosphatidylethanolamines. Some lipoxygenase products from linoleic acid are found in group 7; other lipoxygenase products, several bile acids, and cortisol were found in category 8.

## 4. Discussion

The analysis strategy employed here, combining proteomics with lipidomics, revealed novel and previously unrecognized regulatory events of oxylipins upon LPS challenge. For the induction of inflammation, we used a unique in vivo human model in which healthy individuals are challenged with LPS, allowing for accurate time course recording [1]. The specificity of the proteins upregulated by LPS indicated a temporal sequence of cell types involved as inflammatory players involved: first, the endothelial and platelet-derived protein von Willebrand factor (VWF) was involved [51], followed by the predominantly liver-derived acute phase protein SAA1 and the innate immune cell-derived protein S100A9. Subsequently, the liver-derived inflammatory marker CRP became prominent. Lipidomics revealed more significant events following LPS administration than proteomics but supported the interpretation in terms of the cell types involved. Polyunsaturated fatty acids, significantly induced 2 h after LPS challenge (Appendix A), may be released due to phospholipase A2 activity, previously described to be activated in innate immune cells upon inflammatory activation via sepsis [52]. Trans-fatty acids such as trans-DHA may indicate the chemical effects of typically endothelium-derived nitric oxide [53,54].

Plasma levels of the lysophospholipid sphingosine-1-phosphate are mainly regulated by endothelial cells, erythrocytes, and platelets [55]. Similar to the PUFAs, the downregulation of plasma sphingosine-1-phosphate observed in response to LPS challenge closely mirrors previous findings reported in the context of sepsis [56]. The direct contribution of blood components to the systemic oxylipin response to LPS is also indicated by the upregulation of TxB2, 12-HHTrE, and 12-HETE, as these molecules were previously found to be released from activated platelets [34]. The deregulated bile acids, inhibitors of the NLRP3 inflammasome [57], observed 4 h after the challenge, strongly suggest the involvement of the liver in the modulation of inflammatory processes [58].

A striking observation was the uniform downregulation of numerous lysophospholipids, most prominently LPE (0:0/18:1), LPE (0:0/18:2), LPC (0:0/18:1), and LPC (0:0/18:2) upon LPS challenge (Appendix A). Lysophospholipids are known to be abundant in plasma as a result of the Lands cycle [59], maintaining the generally high turnover of PUFAs within phospholipids [60,61,62]. Phospholipids with unsaturated fatty acids are essential for the function of the endoplasmic reticulum [63] and mitochondria [64] but vulnerable with regard to inflammation-associated oxidation [65]. The present data thus indicate that inflammation-induced oxidation of phospholipids may be almost immediately managed and regenerated via the Lands cycle, apparently resulting in the systemic consumption of PUFA-containing plasma lysophospholipids. Remarkably, besides the liver, erythrocytes have also been described to play an active role in the Lands cycle [66], suggesting that erythrocyte metabolism may be involved in the early response to inflammatory activation in humans. This may also relate to sphingosine-1-phosphate, an important modulator of inflammation [49,61,67] specifically released by platelets, endothelial cells, and erythrocytes [68,69]. Sphingosine-1-phosphate was demonstrated to be released by erythrocytes upon hypoxia, eventually regulating the oxygen supply [70]. Therefore, our results support an early adaptive response of erythrocytes to mitigate tissue hypoxia caused by the inflammation-induced increase in oxygen expenditure [71].

Considering the known functions of the presently observed systemic modulators of inflammation, it is striking that most of them seem to contribute to the resolution of inflammation. While CRP is regarded as pro-inflammatory, it mediates the opsonization of bacteria, immune complexes, and damaged cells, thereby mediating their elimination [72]. Similarly, SAA1 may stimulate pro-inflammatory cytokine expression but also facilitates the removal of membrane debris and toxic lipids formed during inflammation [73]. Antioxidative and anti-inflammatory properties have been attributed to polyunsaturated fatty acids [74]. The role of endocannabinoids regulating immune functions has been recognized rather recently [75]. To understand the pro- and anti-inflammatory properties and physiological roles of lysophospholipids, their fatty acyl composition, production, transport, and utilization mechanisms must be considered in detail [61]. Furthermore, taurocholic acid and glycocholic acid have been described to inhibit inflammation [76].

Evidently, the most potent inflammation-promoting factors, such as cytokines and prostaglandins, were not detected in the present study. This may not be surprising as these pro-inflammatory factors are hardly formed within blood, but rather in the interstitium [77], where they act rather locally [78], are short-lived [79], and are detectable only at very low concentration levels [80]. In contrast, the modulation and resolution of potentially damaging inflammatory responses may be mediated by more long-lived molecules accumulating higher concentration levels, increasing their chance of being positively detected. The present data suggested that the kinetics of formation and degradation of these molecules varies remarkably, making it potentially difficult to obtain significant results in diseased patients with unknown onset of the inflammatory responses. This may also apply to the immune-modulatory effect of the presently applied intervention, which resulted in significant alterations of different molecules at different time points when compared to the placebo group. Without matching the time points presently obtained due to the artificially induced onset of inflammation, no significant effects would have been observed.

This may also explain why only a few studies exist regarding the deregulation of plasma levels of oxylipins and other inflammatory modulators in humans. Many researchers may have observed that the measurements would not be sufficiently robust and precise to support meaningful conclusions. This study demonstrated the successful determination of these molecules in plasma from human individuals, revealing complex regulatory patterns and time courses after a defined challenge. This may indeed have resulted from the synchronous initiation of inflammation, a specific feature of this study. The kinetics observed after LPS challenge were highly reproducible and characteristic for each molecule (Figure 6). As groups of molecules with common precursors or synthesizing enzymes showed rather similar kinetics, they will serve to learn more about inflammatory regulatory mechanisms and the responsible physiological processes. The combination of proteomics and lipidomics allowed us to match the kinetics of formation and thus mutually determine the specific sources of proteins and lipids.

A modulation of marker proteins for inflammatory processes by antioxidant supplementation was actually not observed. This is in line with previous investigations [81] and may suggest that the initiation of the corresponding inflammatory processes was not significantly affected by antioxidants. However, we have observed several oxylipins specifically affected by this intervention (Figure 5). The increase of 20-HEPE and 22-HDoHE in plasma upon supplementation with ω-3 fatty acids has already been observed previously and was thus successfully confirmed by the present data [82]. However, at that time, no inflammatory challenge had been investigated. Here, we also described the downregulation of sphingosine-1-phosphate, a potent pro-inflammatory mediator, by antioxidative intervention. It is plausible to assume that the metabolism of inflammatory mediators may be affected by antioxidants and may thus account for the clinical observations [2,22,83]. Thus, sphingosine-1-phosphate and the anti-inflammatory oxylipins presently affected by antioxidative supplementation represent plausible candidates potentially mediating such effects.

A potential weakness of this study is the lack of formal proof of this hypothesis. Future studies will be required to deconvolute all the synthesis, uptake, degradation, and transport processes that make up the extraordinarily complex regulatory system of inflammation involving proteins and lipids. Another limitation is the lack of absolute quantification, which is still difficult to achieve for most oxylipins due to the lack of analytical standards. However, the future realization of absolute quantification methods may exploit the clock-like properties of oxylipin variations to support the deconvolution of the responsible physiological processes, potentially leading to descriptions of patient states relevant for clinical assessment.

## 5. Conclusions

The present data demonstrate that lipid mediators, rather than proteins, may be responsible for the improvement in clinical symptoms observed with antioxidant supplementation. We describe, for the first time, complex pattern of lipid mediator alterations in response to inflammatory challenge, which may stimulate interest in the use of oxylipins as biomarkers in the future. Remarkably, the time-course analyses indicated that erythrocytes and platelets, which have hardly been previously considered as relevant entities in the regulation of inflammation, may be the main targets of the antioxidant supplementation and may contribute to the resolution of inflammation.

## Figures and Tables

**Figure 1 antioxidants-14-00536-f001:**
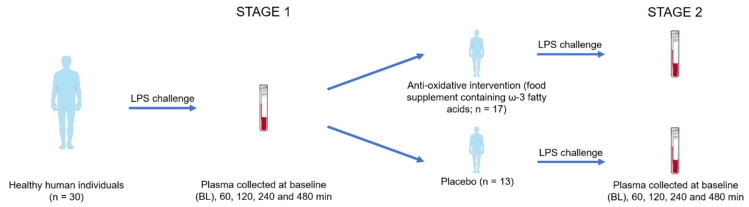
Study design for the investigation of proteome and oxylipin alterations accompanying LPS challenge at stage 1 and the effects of an antioxidative intervention at stage 2. The figure was partly generated using Servier Medical Art, provided by Servier, licensed under a Creative Commons Attribution 3.0 Unported license.

**Figure 2 antioxidants-14-00536-f002:**
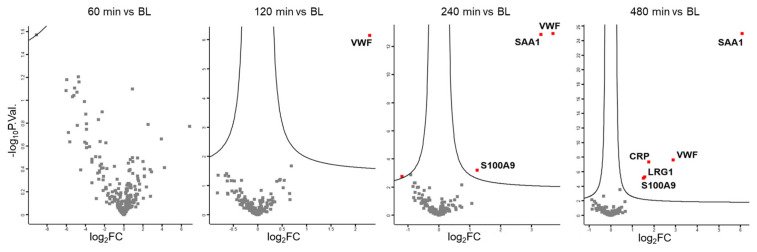
Volcano plots displaying proteome alterations after LPS challenge (stage 1) demonstrating the involvement of platelets (VWF), innate immune cells (S100A9), and the liver (SAA1, CRP, LRG1).

**Figure 3 antioxidants-14-00536-f003:**
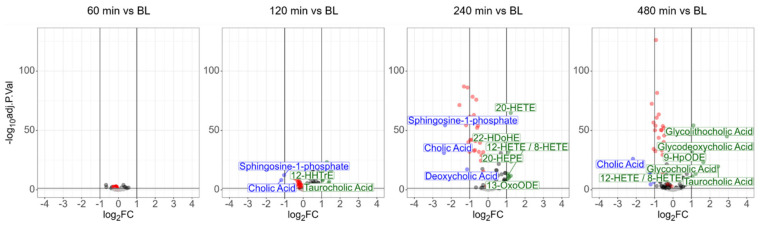
Oxylipins, lysophospholipids (marked as red dots), bile acids, and sphingosine-1-phosphate are significantly deregulated upon LPS challenge.

**Figure 4 antioxidants-14-00536-f004:**
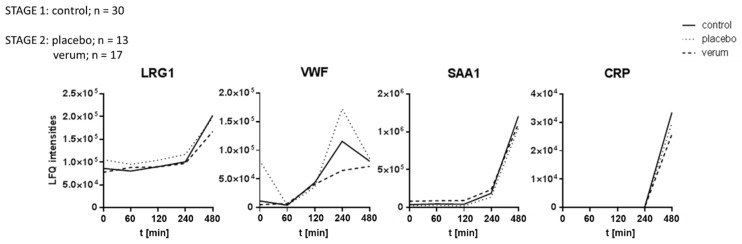
Time course of average LFQ values obtained for LRG1, VWF, SAA1, and CRP. No significant proteome alterations were observed in the verum group treated with antioxidants compared to the placebo group.

**Figure 5 antioxidants-14-00536-f005:**
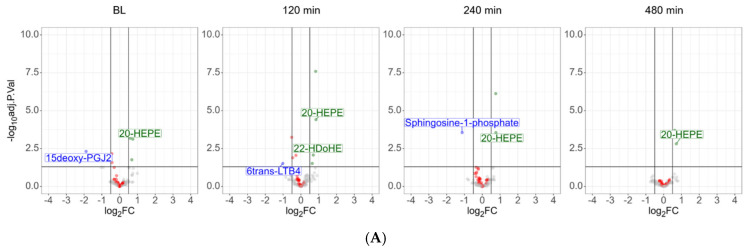
The antioxidant intervention affected the levels of several lipid mediators. (**A**) Volcano plots depicting differences between the verum group and the placebo group. Lysolipids are marked in red. (**B**) Detailed depiction of the individual measurement results of selected lipid mediators at stage 1 and stage 2; analyte levels of the verum group are marked in red; analyte levels of the placebo group are marked in blue. * adj. *p*-value ≤ 0.05.

**Figure 6 antioxidants-14-00536-f006:**
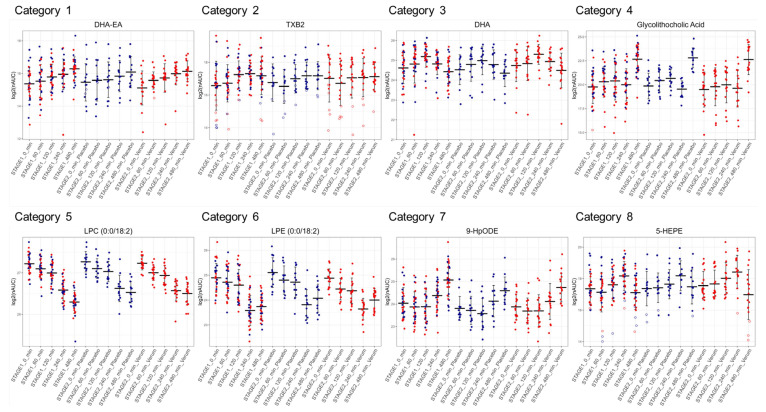
The plasma lipids deregulated upon LPS challenge show distinct kinetic properties. The log2-transformed normalized area under the curve values (nAUC) of the corresponding oxylipins are depicted for each donor including mean values at all time points after LPS challenge at stage 1 (the participants are already marked in color according to their affiliation at stage 2), at stage 2 (placebo group, marked in blue), and at stage 2 verum group (marked in red). One representative oxylipin of each group was chosen to display the different kinetic categories: (1) continuously up; (2) first up, then partially down; (3) first up, then down below baseline levels; (4) up, down, and up again; (5) continuously down; (6) first down then partially up; (7) first down, then up above baseline levels; (8) down, up and down again. Note that the three time series are experimentally entirely independent of each other.

## Data Availability

The mass spectrometry proteomics data have been deposited to the Proteo-meXchange Consortium via the PRIDE [44] partner repository with the dataset identifier PXD051013. Data derived from the lipid mediator analysis are available at the NIH Common Fund’s National Metabolomics Data Repository (NMDR) website, the Metabolomics Workbench, https://www.metabolomicsworkbench.org (accessed on 2 February 2025) [45], where they have been assigned to the following studies: Study ID ST003137 and ST003138. The DOI for this project (PR001950) is: http://dx.doi.org/10.21228/M8NX50.

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
