# Peer review of "Time Course of Plasma Proteomic and Oxylipin Changes Induced by LPS Challenge and Modulated by Antioxidant Supplementation in a Randomized Controlled Trial"

_antioxidants, 2025, doi:10.3390/antiox14050536_

Round 1
Reviewer 1 Report
Methods
Supplementary Table 1: Beta-carotene is listed in the table as not being used under any condition, yet it appears in the table. This seems to have been extracted from the experimental data of a previous study.
Results:
Line 234: vWF is primarily produced and released by endothelial cells, stored in platelets, and also produced in other tissues. How can the authors directly interpret the increase in vWF is derived from endothelial- and platelet-derived wVF?
Line 271: Sphingosine-1-phosphate (S1P) is produced in endothelial cells, not just RBCs. However, the authors describe S1P as a product specific to erythrocytes and interpret the inflammation response as mediated through erythrocytes. I believe the gap between the data and this interpretation is large for such a conclusion to be drawn.
.
Author Response
The title seems somewhat ambiguous, making it unclear what the primary focus of the study is. It would be helpful to incorporate key experimental terms, such as "oxylipin alterations in blood plasma" and "study design," to make the research content more specific and clearer. A revised title like "Molecular Alterations in Blood Plasma: Proteomic and Oxylipin Changes Induced by Antioxidative Supplementation in a Randomized Controlled Trial" would provide a better understanding of the article's focus and may be more appealing to readers.
Answer: We thank the reviewer for this helpful comment and revised the title accordingly, which now is: "Time Course of Plasma Proteomic and Oxylipin Changes Induced by a LPS Challenge and Modulated by Antioxidative Supplementation in a Randomized Controlled Trial"
It would be helpful if the authors could provide a more detailed explanation of the differences between this study and previous research in the field.
Answer: We appreciate this helpful comment. Actually we have reformulated the Introduction quite substantially to better communicate the previous research and the aims of this study.
Clinical Design: To induce systemic inflammation and oxidative stress, an intravenous infusion of a bolus containing 2 ng/kg LPS was used. → It is surprising that the study was approved by the IRB given that LPS, a potent immune activator, was administered intravenously, followed by blood draws five times, and this process was repeated after a secondary visit with additional LPS administration and five more blood draws. Were participants compensated beyond transport costs for their involvement in this clinical trial? LPS is a strong immune activator, and even at low doses, the risk of side effects such as hypersensitivity cannot be ruled out. Please provide references from other studies (no the authors previous studies) that show no significant side effects at this dosage, as indicated by previous human trials. Additionally, the study protocol mentions that it was approved by the Ethics Committee of the Medical University of Vienna (EC No.: 64/2009). Since this approval was for a different study, I am concerned whether this IRB approval can cover the current experiment. Were new volunteers recruited for this study, or were existing samples used? If the latter, were there any concerns about sample preservation?
Answer: We apologize that we have missed to explain that the study was actually registered at ClinicalTrials.gov number NCT00914576. Here, all the requested information can be retrieved, and for sure this trial is compliant with the most strict international regulations. This important information is now contained in the manuscript. We have indeed applied a well-established protocol, which was approved by our and other Ethics Commitees many times as cited in the Introduction.
Supplementary Table 1: Beta-carotene is listed in the table as not being used under any condition, yet it appears in the table. This seems to have been extracted from the experimental data of a previous study.
Answer: This was only to specifiy that no Beta-carotene was included in contrast to similar intervention designs. We have no updated the file accordingly.
Results: Line 234: vWF is primarily produced and released by endothelial cells, stored in platelets, and also produced in other tissues. How can the authors directly interpret the increase in vWF is derived from endothelial- and platelet-derived wVF?
Answer: The expression specificity of vWF specifies that information as indicated in the cited paper (Refs 46, 47).
Line 271: Sphingosine-1-phosphate (S1P) is produced in endothelial cells, not just RBCs. However, the authors describe S1P as a product specific to erythrocytes and interpret the inflammation response as mediated through erythrocytes. I believe the gap between the data and this interpretation is large for such a conclusion to be drawn.
Answer: It is true that S1P is also produced by endothelial cells, as we have now clearly stated in the manuscript including the Abstract. In general, we followed the arguments and data as published in recognized Journals.
Methods
Supplementary Table 1: Beta-carotene is listed in the table as not being used under any condition, yet it appears in the table. This seems to have been extracted from the experimental data of a previous study.
Answer: As outlined above, this error was corrected.
Results:
Line 234: vWF is primarily produced and released by endothelial cells, stored in platelets, and also produced in other tissues. How can the authors directly interpret the increase in vWF is derived from endothelial- and platelet-derived wVF?
Answer: As outlined above, we have corrected this comprehensively.
Line 271: Sphingosine-1-phosphate (S1P) is produced in endothelial cells, not just RBCs. However, the authors describe S1P as a product specific to erythrocytes and interpret the inflammation response as mediated through erythrocytes. I believe the gap between the data and this interpretation is large for such a conclusion to be drawn.
Answer: We have now corrected this statement accordingly.
Reviewer 2 Report
Dear Authors,
The article titled "Antioxidative supplementation has minimal impact on the initiation of inflammation but may instead facilitate its resolution" is an excellent work that elucidates the initial mechanisms of inflammation induced by LPS in humans. Furthermore, the study employs an excellent and refined methodology in lipidomics and proteomics, which allows us to further refine molecular mechanisms and explore new perspectives on markers associated with acute inflammation.
Although no conclusive effects of antioxidant supplementation after 14 days of intervention were observed in the the study, it allows us to consider some interesting possibilities in the pro-inflammatory mechanisms induced by LPS.
And some points to discuss is related below.
Some key questions regarding the conduct and randomization of the study groups.
Point 1: It is necessary to mention whether the participants experienced any adverse effects after the administration of LPS, placebo or omega-3 supplementation.
Point 2: It is necessary to further describe whether any randomization software was used for the selected groups and why the placebo group had 13 participants while the treatment group had 17.
Point 3: Other questions include whether there was any desensitization after stage 1 to start stage 2.
And Point 4: how was the control over the participants' diet managed in the study? Was the diet provided by the researchers or was it free choice of each individual? Was there any intervention or guidance on dietary intake provided by a dietitian? Was calculated type or amount of fatty acids consumption, for example?
Author Response
We would like to thank the reviewer for the positive and helpful comments. Please find enclosed our point to point answers:
Some key questions regarding the conduct and randomization of the study groups.
Point 1: It is necessary to mention whether the participants experienced any adverse effects after the administration of LPS, placebo or omega-3 supplementation.
Answer: Actually this study was registered at ClinicalTrials.gov number NCT00914576, and all clinical details can be retrieved from there. No adverse effects of the challenge were reported other than the announced effects including symptoms reminiscent to a light viral infection.
Point 2: It is necessary to further describe whether any randomization software was used for the selected groups and why the placebo group had 13 participants while the treatment group had 17.
Answer: As outlined above, this study followed strict rules and was registered at ClinicalTrials.gov . The different numbers were consequent to individuals not showing up as planned.
Point 3: Other questions include whether there was any desensitization after stage 1 to start stage 2.
Answer: No, there was no specific desensitization. As demonstrated by the data, there was no memory effect recorded whatsoever.
And Point 4: how was the control over the participants' diet managed in the study? Was the diet provided by the researchers or was it free choice of each individual? Was there any intervention or guidance on dietary intake provided by a dietitian? Was calculated type or amount of fatty acids consumption, for example?
Answer: Thank you for this excellent question. As recorded in the study design accessible via ClinicalTrials.gov, the participants indeed all got the same meal served by the clinics during the intervention, as they spent the whole time period in the clinics under full supervision.
Reviewer 3 Report
To study the impact of antioxidative intervention in LPS induced inflammation, the authors used a very good controlled study model that should also be used in further studies. They found evidence that the omega-3 fatty acid treatment primarily improves resolution of inflammation rather than its induction. Taking into acount existing literature, however, the novelty of the findings is limited.
The manuscript could be much improved if data on cytokines and prostaglandins were included (which had been done in related similar studies); because of their low levels this is likely not possible by the methods applied in this paper. The authors may analyze proband samples using ELISA or other suitable methods.
Line 412ff: The authors discuss that they could not detect cytokines and prostaglandins. The authors should discuss more that this, as I assume, due to limitations of the used method.
Line 445-446: As pointed out by the authors, the increase in 20-HEPE and 22-HDoHE after treatment with omega-3 fatty acids had been observed in an earlier study (Fischer et al. (2014) J Lipid Res 55:1150); Thus, the present study confirms the study by Fischer et al. (not vice versa, as the authors stated in their discussion).
Author Response
We would like to thank the reviewer for the positive and helpful comments. Please find enclosed our point to point answer:
To study the impact of antioxidative intervention in LPS induced inflammation, the authors used a very good controlled study model that should also be used in further studies. They found evidence that the omega-3 fatty acid treatment primarily improves resolution of inflammation rather than its induction. Taking into acount existing literature, however, the novelty of the findings is limited.
Answer: We rewrote the Introduction in order to better present what data already existed and what was new. In addition, we have now included a limitations paragraph and a Conclusions paragraph, which may help to better understand the implications of the novel findings.
The manuscript could be much improved if data on cytokines and prostaglandins were included (which had been done in related similar studies); because of their low levels this is likely not possible by the methods applied in this paper. The authors may analyze proband samples using ELISA or other suitable methods.
Answer: Thank you for this comment. We have now included additional explanations why cytokine measurements are hardly performed and included a study which has determined cytokine expression profiles via RT-PCR in the same clinical trial settings. It is important to note that the exact reproduction of the kinetics of CRP strongly indicates that there is no effect of the antioxidant supplementation on cytokine release, as any such change would affect CRP.
Line 412ff: The authors discuss that they could not detect cytokines and prostaglandins. The authors should discuss more that this, as I assume, due to limitations of the used method.
Answer: Thank you for this comment. Yes we have detailed the implictions in the Introduction accordingly.
Line 445-446: As pointed out by the authors, the increase in 20-HEPE and 22-HDoHE after treatment with omega-3 fatty acids had been observed in an earlier study (Fischer et al. (2014) J Lipid Res 55:1150); Thus, the present study confirms the study by Fischer et al. (not vice versa, as the authors stated in their discussion).
Answer: We thank the reviewer for this helpful comment. We have corrected this statement accordingly.
Round 2
Reviewer 1 Report
.
.
Author Response
We thank the reviewer for the helpful comments and adjusted the conclusion accordingly.
Reviewer 3 Report
The authors have positively adressed all commenst of this reviewer and significantly improved the manuscript.
The introduction is very much improved. Figures are much better readable now. Critical points of the first review have been adressed in a satisfactory manner.
Author Response
We thank the reviewer for the positive feedback!